# Prediction of Response and Survival Following Treatment with Azacitidine for Relapse of Acute Myeloid Leukemia and Myelodysplastic Syndromes after Allogeneic Hematopoietic Stem Cell Transplantation

**DOI:** 10.3390/cancers12082255

**Published:** 2020-08-12

**Authors:** Christina Rautenberg, Anika Bergmann, Ulrich Germing, Caroline Fischermanns, Sabrina Pechtel, Jennifer Kaivers, Paul Jäger, Esther Schuler, Rainer Haas, Guido Kobbe, Thomas Schroeder

**Affiliations:** Department of Hematology, Oncology and Clinical Immunology, University Hospital Duesseldorf, Medical Faculty, Heinrich Heine—University, Duesseldorf, Moorenstr. 5, 40225 Duesseldorf, Germany; christina.rautenberg@med.uni-duesseldorf.de (C.R.); anber126@hhu.de (A.B.); germing@med.uni-duesseldorf.de (U.G.); carolin.fischermanns@med.uni-duesseldorf.de (C.F.); Sabrina.pechtel@med.uni-duesseldorf.de (S.P.); Jennifer.Kaivers@med.uni-duesseldorf.de (J.K.); PaulSebastian.Jaeger@med.uni-duesseldorf.de (P.J.); esther.schuler@med.uni-duesseldorf.de (E.S.); Haas@med.uni-duesseldorf.de (R.H.); Kobbe@med.uni-duesseldorf.de (G.K.)

**Keywords:** AML, MDS, allogeneic stem cell transplantation, relapse, azacitidine

## Abstract

To provide long-term outcome data and predictors for response and survival, we retrospectively analyzed all 151 patients with relapse of myeloid neoplasms after allogeneic hematopoietic stem cell transplantation (allo-HSCT) who were uniformly treated with first-line azacitidine (Aza) salvage therapy at our center. Patients were treated for molecular (39%) or hematologic relapse (61%), with a median of 5 cycles of Aza and at least one donor lymphocyte infusion in 70% of patients. Overall response was 46%, with 41% achieving complete (CR) and 5% achieving partial remission. CR was achieved after a median of 4 cycles and lasted for a median of 11 months (range 0.9 to 120 months). With a median follow-up of 22 months (range: 1 to 122 months), the 2-year survival rate was 38% ± 9%, including 17 patients with ongoing remission for >5 years. Based on results from multivariate analyses, molecular relapse and time to relapse were integrated into a score, clearly dividing patients into 3 subgroups with CR rates of 71%, 39%, and 29%; and 2-year survival rates of 64%, 38%, and 27%, respectively. In the subgroup of MDS and secondary AML, receiving upfront transplantation was associated with superior response and survival, and therefore pretransplant strategy was integrated together with relapse type into a MDS–sAML-specific score. Overall, Aza enables meaningful responses and long-term survival, which is a predictable with a simple-to-use scoring system.

## 1. Introduction

During the last decade, the hypomethylating agents (HMA) azacitidine (Aza) and decitabine (DAC) have become a clinically relevant treatment option for patients with relapse of acute myeloid leukemia (AML) and myelodysplastic syndromes (MDS) after allogeneic hematopoietic stem cell transplantation (allo-HSCT) [1,2,3,4,5]. This was based on several retrospective analyses, as well as a few prospective single-arm trials reporting response rates ranging from 10% to 35% and 2-year overall survival ranging from 12% to 29% at 2 years [6,7,8,9,10,11,12]. Two of these analyses identified diagnosis of MDS instead of AML and disease burden at relapse as predictors for response and survival [7,11], while results regarding the interval between transplant and relapse were controversial. This controversy and limited knowledge about predictive factors probably results in part from limited data access in these registry-based analyses, as well as different inclusion criteria, with one analysis including molecular relapses and one only focusing on hematologic relapses. The latter aspect is of relevance, since during the last years physicians routinely employ available techniques to monitor minimal residual disease (MRD), aiming to detect imminent relapse and to start preemptive therapy at the stage of low disease burden [13,14]. Furthermore, molecular characterization of AML and MDS has become a standard for risk stratification and guidance of treatment decisions in the first-line setting, but may also impact response and survival in case of relapse after transplant. Finally, in patients with MDS there is controversy among experts regarding the value of pretransplant cytoreduction by intensive chemotherapy and hypomethylating agents [15,16]. Preliminary data from our group suggested that in contrast to upfront transplantation, pretransplant cytoreduction may negatively impact response and survival to HMA in case of relapse after allo-HSCT [17]. To address these issues and to expand the knowledge about predictive factors, we analyzed data from a well-annotated cohort of 151 patients with a long follow-up, including a relevant fraction of patients with molecular relapse, treated with Aza for relapse after allo-HSCT at our center. This analysis aims to provide practical information for physicians, supporting their selection of optimal treatment and counseling of their patients in this challenging situation.

## 2. Results

### 2.1. Patient Characteristics

We retrospectively analyzed data from 151 patients (median age: 54 years, range 19 to 71 years) with relapse of AML (n = 90, 60%), MDS (n = 49, 32%) or myeloproliferative neoplasm (n = 12, 8%) after allo-HSCT. Prior to transplant, 43 patients (28%) did not receive cytoreductive treatment (upfront transplantation), whereas the majority of patients (n = 104, 69%, missing information n = 4) either received intensive CTX (n = 90, 87%) or HMA (n = 14, 13%). Of the latter, which were summarized as the treatment group for the sake of this analysis, 50 patients (48%) were transplanted in 1st (n = 45) or 2nd (n = 5) CR, while 54 patients (52%) were not in remission at transplant, either having not achieved complete remission (n = 34) or having relapsed (n = 20). The majority of patients (n = 97, 64%) had received a reduced intensity conditioning (RIC), whereas 54 patients (36%) underwent allo-HSCT after standard-dose conditioning (15). Relapse occurred in median 147 days (range, 28 to 6440 days) after allo-HSCT. Fifty-nine patients (39%) experienced molecular relapse, whereas 92 patients (61%) suffered from hematologic relapse. Molecular relapse in these 59 patients was detected only by reoccurence of molecular (n = 25) or cytogenetic (n = 6) abnormalities, decrease of donor chimerism (n = 6), sex-mismatched donor-recipient constellation in XY-FISH (n = 1), or a combination of more than one applicable MRD method (n = 21). Five patients developed extramedullary relapse (2 patients with skin infiltrations, 3 patients with meningeosis leukemica), which was associated with concurrent molecular (n = 1) or hematologic relapse (n = 4). Extended information regarding patient, transplant, and relapse characteristics are summarized in Table 1 and Table 2 and Appendix A.

### 2.2. Treatment

Aza was the first salvage therapy for relapse in all 151 patients, as reflected by a median time between diagnosis of relapse and initiation of Aza treatment of 10 days (range, 0 to 91 days). Patients received a median of 5 Aza cycles (range 1 to 18), with 145 patients treated with 75 mg/m^2^ per day for 7 days and 6 patients treated with a Aza dosage of 100 mg/m^2^ for 5 days according to the protocol of the prospective trial (NCT00795548) [12]. Five patients (3%) with FMS-related tyrosine kinase 3—internal tandem duplication (FLT3-ITD)-mutated AML additionally received off-label treatment with Sorafenib in a dosage of 2 × 400 mg orally per day [25]. Both dosing regimens comprised a 28-day schedule and we envisaged the administration of 6 to 8 cycles of Aza. Administration of DLI was planned after every second Aza cycle, depending on individual response and tolerability. Overall, 105 patients (70%) received at least one DLI, whereas in the remaining 46 patients (30%) no DLIs were administered. The most common reasons to surrender the administration of DLI were progressive disease (n = 20, 43%) and GvHD (n = 11, 24%). Other reasons included donor unavailability, concerns due to previous haploidentical transplantation, poor performance status, and envisaged second transplant in two patients (missing information in 4 patients). Median time to 1st DLI was 55 days (range, 5 to 366) from the commencement of Aza. Median number of DLIs per patient was 2 (range, 1–6), corresponding to a median number of CD3+ T-cells per patient of 6 × 10^6^/kg bodyweight (range, 0.5 to 116 × 10^6^/kg bodyweight).

### 2.3. Response and Outcome

Following this treatment, 62 patients (41%) achieved CR and another 7 patients (5%) achieved PR, resulting in an overall response rate (ORR) of 46%, with no significant differences between the two different Aza dosing schemes (data not shown). Median time to achievement of CR was 107 days (range, 16 to 349), corresponding to a median of 4 cycles of Aza (range, 1–9). Thirty-six patients (58%) remained in ongoing CR for a median of 22 months (range, 1 to 120 months), whereas 26 patients (42%) relapsed again after a median of 5 months (range, 1 to 76 months). The majority of patients (n = 53, 85%) achieving a CR had received DLI, with a median number of 3 DLIs (range, 1 to 5) per patient. Of these, 9 patients (17%) were already in remission before application of DLI, whereas the majority (n = 44, 83%) achieved remission after administration of the first DLI. Nine patients (15%) with remission did not receive any DLI. Reasons for not administering DLI were GvHD (n = 6) and envisaged second transplant (n = 1) (missing information in 2 patients). Of these 9 patients, 7 patients remained in ongoing remission for a median of 653 days (range, 122–1687), but 2 patients experienced another relapse after 57 and 64 days. After a median follow-up of 22 months (range, 1 to 122 months), the 2-year OS rate of the entire cohort was 38% ± 9% (Figure 1). Seventy-four patients (49%) were alive at last follow-up, whereas 77 patients (51%) had died. Causes of death were underlying disease, cytopenia-related infections, bleeding, GvHD, or other reasons in 62 (81%), 12 (16%), 1 (1%), 1 (1%), and 1 (1%) patients, respectively. Due to Aza failure 17 patients (11%) proceeded to second transplant. Of these, 14 patients subsequently died (2 non-relapse mortality, 12 disease-related), 3 patients are alive, and 2 are in molecular remission for 38 and 50 months after second transplant, whereas one patient experienced another molecular relapse after 11.5 months. Trying to address the potential role of DLI, we performed a landmark analysis including only those patients who were alive ≥ 55 days following Aza-based salvage therapy, as this time point represents the median time to first DLI. Here, patients receiving DLI showed a significantly longer OS compared to those patients who did not (*p* = 0.04, HR 1.898, 95% CI 1.008–3.200; Appendix A).

### 2.4. Predictors for Response and Overall Survival

A major scope of this study was to identify predictive factors for response and survival of patients treated with Aza and DLI in order to support physicians during the decision-making processes. Thus, when focusing on the entire cohort in univariate analysis, disease burden in terms of molecular relapse and a bone marrow blast count at relapse of ≤the median of 7%, as well as an interval of ≥6 months between allo-HSCT and relapse, were identified as predictors for achievement of complete remission. Of these, only molecular-only relapse (*p* = 0.0001) was confirmed as significant predictor for response to Azacitidine in multivariate analysis (Table 3). Furthermore, in univariate analysis diagnosis of MDS compared to sAML and de novo AML, a molecular instead of frank hematological relapse, a bone marrow blast count at relapse of ≤the median of 7%, as well as an interval of ≥6 months between allo-HSCT and relapse, were identified as predictors for OS. In multivariate analysis, molecular relapse (*p* = 0.002) and an interval of ≥6 months between allo-HSCT and diagnosis of relapse (*p* = 0.026) retained their favorable impacts on OS (Table 3). 

The issue of whether pretransplant cytoreduction should be performed in patients with advanced MDS and sAML is a matter of an ongoing debate. Furthermore, preliminary data suggest that pretransplant strategy (cytoreduction versus upfront transplantation) in these patients may also influence response and outcome following treatment with Aza and DLI in case of relapse [14]. To address this, we focused on this specific subgroup of patients with MDS (n = 49) and sAML (n = 17) with a BM blast count ranging from 20% to 29% (formerly RAEB-T) and performed additional uni- and multivariate analyses regarding response to and outcome after salvage therapy with Aza. As such, upfront transplantation and disease burden at relapse reflected by molecular-only relapse and BM blast count ≤ the median of 8% at relapse in this subgroup were associated with a higher likelihood to achieve CR in univariate analysis, whereas multivariate analysis confirmed the prognostic impact of upfront transplantation (*p* = 0.006) and molecular-only relapse (*p* = 0.002), as well as male gender (*p* = 0.003) (Appendix A). Furthermore, in addition to molecular-only relapse we identified BM infiltration at relapse ≤ median of 8% blasts, an interval of ≥6 months between allo-HSCT and relapse, and upfront transplantation instead of pretransplant cytoreduction as predictors for prolonged OS. In multivariate analysis, we confirmed upfront transplantation (*p* = 0.009), BM blast count ≤ 8% at relapse (*p* = 0.001), and absence of complex karyotype (*p* = 0.021) as favorable predictors for OS (Appendix A).

### 2.5. Aza Prognostic Scoring System—For Relapse after Transplant (APSS-R)

In the next step, we aimed for development of a practical and simple-to-use risk score that allows physicians to predict response and survival of their patients after Aza-based salvage therapy. This system is based on the results derived from multivariate analyses and assigns points to predictive factors according to their relative statistical weight. For the score developed for the entire cohort, the factors relapse type and time to relapse were incorporated. One point was assigned for molecular relapse, while 2 points were given for hematological relapse. For an interval between transplant and relapse ≥ 6 months 0 points were assigned, while 1 point was given for relapse within the first 6 months. As such, the model enables the stratification of patients into 3 risk groups (favorable = 1 point, intermediate = 2 points, unfavorable = 3 points) with clearly differing responses (*p* = 0.0007) and survival rates (*p* = 0.0012) (Figure 2, Table 4). Taking into account the fact that predictive factors in the subgroup of MDS and sAML partially differed, a separate score was developed for this subgroup. To keep the score as simple and practical as the score for the entire cohort, we included type of relapse (molecular 1 point, hematologic 2 points) and pretransplant strategy (upfront 0 point, treated 1 point) into the MDS–sAML-specific score. As such, the score clearly separated patients into three risk groups with distinct responses (*p* = 0.0009) to and survival (*p* = 0.0023) after Aza-based salvage therapy (Figure 3 and Table 5).

## 3. Discussion

In the present study, we analyzed data from all 151 patients consecutively and uniformly treated with Aza as first salvage therapy for relapsed myeloid neoplasms after allo-HSCT at our center from 2005 until 2018. This well-annotated data set and long-term follow-up of these patients enabled us to provide robust response and survival rates, including a relevant proportion of patients preemptively treated at the stage of molecular relapse. It also gave us the opportunity to identify predictors for response and survival and to integrate them into two practical, easy-to-use scores to predict outcome after Aza-based salvage therapy.

Previous reports on Aza-based salvage therapy for relapse after allo-HSCT were derived from a prospective trial [12] and retrospective single- [9,10], multicenter [7] and registry-based [11] analyses, which covered 30 to 181 patients. Patients in these reports were heterogeneous with regard to previous lines of relapse therapy prior to Aza, the use of DLI, and suffering from frank hematological relapse [9,10,11,12] in all but one analysis, which included 12% of patients with molecular relapse [7]. In these reports, CR rates and 2-year OS rates ranged from 10% to 35% and 12% to 29%, respectively. In contrast, our cohort reported here solely included patients that received Aza as first intervention and contained a relevant proportion of patients (39%) preemptively treated for pending relapse. This strategy to regularly monitor MRD and to preemptively initiate a therapeutic intervention when MRD is detected nowadays is the standard [13,26,27]. Indeed, this has mainly contributed to a CR rate of 41% and 2-year OS rate of 38%, including 17 patients alive in ongoing remission without any further anti-leukemic treatment for >5 years (up to 10 years). Overall, these results appear to be not only favorable when compared to the previous reports on the use of Aza [9,10,11,12,17], but also when considering outcome after alternative approaches, such as intensive chemotherapy or second transplantation [28,29,30]. Still, the retrospective nature of these indirect comparisons and potential selection bias have to be taken into account. 

In this challenging situation of relapse after allo-HSCT, physicians need to carefully weigh and select the most promising treatment for the individual patient. In previous work, including one study from our group, disease burden at the time of relapse had already been identified as a predictor for response and OS [7,11]. By initially looking at the entire cohort, we here confirm this finding, showing that a low disease burden reflected by molecular relapse is a predictor for response and survival. Furthermore, our finding that a short interval between transplant and relapse functions as a negative predictor for OS is in line with the results from Craddock et al. In contrast, we could not corroborate the previous finding that the diagnosis of MDS is associated with a higher likelihood of response and survival after Aza-based salvage therapy [7,11]. The latter finding supports the idea that independent from the specific underlying myeloid neoplasm (MDS or AML), early detection of disease activity at the lowest level possible and immediate start of Aza are essential prerequisites for treatment success. Aiming to support physicians in this decision making process in the future, we developed a practical score for the entire cohort, which comprises the two variables type of relapse and time to relapse based on their weights in multivariate analysis. This score clearly separated patients into 3 groups with different outcomes: patients in groups 1 and 2, e.g., those with late molecular relapse who had promising CR and 2-year OS rates of 71% and 64%, respectively; patients in group 3 with early hematologic relapse exhibited CR and 2-year OS rates of 29% and 27%, suggesting that in these patients alternative strategies such as addition of a second compound (i.e., Lenalidomide [31,32], Venetoclax [33]) or investigational agents might improve their prognosis. Furthermore, the currently available FLT3 inhibitors such as Gilteritinib or Sorafenib may represent alternatives, either alone or in combination with Aza, for patients with FLT3 mutation at relapse, which was the case in 11 of 38 of our patients tested at relapse [25,34,35]. A previously reported score was developed exclusively in patients with hematological relapse and incorporated the interval between transplant and relapse and the BM blast count at relapse [11]. In our cohort, in which 39% of patients were treated at the stage of molecular relapse, this score was not able to be separated into three distinct prognostic groups and only had limited predictive impact, as shown in Appendix A and Appendix A. Thus, in times where initiation of preemptive therapy at the lowest detectable evidence level of disease is the major goal, our score based on molecular stage of and time to relapse represents a more practical and easier-to-use tool to support physicians in their choice of therapy and counseling of their patients. 

Acknowledging differences in disease biology between de novo AML and MDS/sAML [36], as well as in the use of pretransplant strategies (upfront transplantation vs. pretransplant cytoreduction), we performed a separate analysis in patients with MDS and sAML (<30% BM blasts). In addition to disease burden at relapse, pretransplant strategy was identified as a predictive factor in multivariate analysis for those patients who were directly transplanted (upfront group), who had significantly higher response and survival rates following Aza-based salvage therapy. This expands findings from our previous analysis [17] and supports a concept of upfront transplantation after sequential conditioning in advanced MDS whenever feasible to potentially avoid the selection of resistant clones, which can be augmented by MRD-guided interventions with Aza and DLI in case of pending relapse. We are aware that this idea remains hypothetical at this point and requires experimental confirmation, including longitudinal sampling during the disease course, ideally in a prospective trial. Consequently, we built a separate score comprising type of relapse and pretransplant strategy for patients with MDS/sAML. As complex karyotype was only a significant predictor for survival but not for response in the subgroup of MDS/sAML patients or in the entire cohort, we did not include this variable in the MDS/sAML-specific score, aiming to provide a practical, simple-to-use score. Furthermore, we did not include gender in the score, as the finding that males responded better to Aza and DLI remains unexplained at this point and requires confirmation. Nevertheless, this two variable-based MDS/sAML-specific scores also clearly divided patients into 3 prognostic subgroups, similarly to the score for the entire group. 

In conclusion, our analysis shows that the combination of Aza and DLI as the first salvage approach enables meaningful response rates and long-term survival in a relevant fraction of patients, which can be easily predicted by the scores provided here. Overall, patients receiving treatment early at the stage of low disease burden especially seem to benefit, while in patients with MDS/sAML the pretransplant strategy also influences the success of post-transplant Aza-based salvage therapy.

## 4. Materials and Methods

### 4.1. Study Design

This retrospective analysis includes 151 patients with AML, MDS, and MPN, who were treated with Aza and envisaged donor lymphocyte infusions (DLI) as first salvage therapy for relapse after allo-HSCT at our center between 2005 and 2019. Data for 66 patients, including 6 patients treated within prospective phase II trials (NCT00795548), have been reported previously [7,12], but were updated for this analysis. The analysis was conducted with the approval of the institutional review board (approval number 2018-331-FmB) and written informed consent was obtained from all patients. 

### 4.2. Definitions and Response Criteria

Hematologic relapse was defined as ≥5% BM blasts, detection of blasts in PB, presence of dysplastic features fulfilling diagnosis criteria for MDS, and extramedullary disease. Molecular relapse was defined as a decrease of donor chimerism to <95% measured in unsorted BM mononuclear cells, evidence of a mixed XY-FISH of ≥4% residual recipient cells, a reoccurrence of known mutations in at least 1% of reads, presence of disease-specific cytogenetic aberrations, or an increase in PB *WT1*-mRNA level > 50 copies per 10^4^
*ABL* copies [37,38] in the absence of any criteria defining hematologic relapse. BM biopsy for response evaluation, including patient-specific MRD-monitoring and BM chimerism analysis, was routinely performed in all of the 151 patients on days +28, +60, +100, +200, and d+300 after allo-HSCT. In case of any deviation from a typical clinical course, BM biopsy was scheduled earlier at the discretion of the treating physician. Cytogenetics, remission status prior to transplant, conditioning intensity, and graft-versus-host disease (GvHD) were defined as previously described [19,20,21,23,24,39]. Regarding response evaluation after relapse, no complete hematologic recovery was required for definition of complete remission (CR), as cytopenia might be related to other factors than the underlying disease, for example GvHD or viral infections. However, for diagnosis of CR, a restoration of complete donor chimerism and negativity of disease specific markers (if applicable) were required. Partial remission (PR) was defined as reduction of bone marrow blast percentage to 5–25% and a decrease of pretreatment bone marrow blast percentage by at least 50%. Outside the prospective trial [12], response monitoring was performed as part of routine care at the discretion of the treating physician. Time to response was calculated from start of Aza treatment until detection of best response, whereas duration of response was defined as the interval between best response and loss of response. OS was calculated as the interval between start of Aza treatment and death or date of last follow-up in surviving patients. Patients that were alive with ongoing remission after HMAs were censored at last follow-up, whereas patients receiving a second allo-HSCT were censored at that date.

### 4.3. Statistical Analyses 

Time-to-event curves were calculated by employing the Kaplan–Meier method and log-rank tests was applied for univariate comparisons. For continuous variables, medians (ranges) were given and the Mann–Whitney test was employed for detection of differences, while for categorical variables frequencies were displayed and differences were estimated using cross tabulation and Fisher’s exact t-test. Variables influencing OS or achievement of response in univariate analysis with a *p*-value < 0.1 were included into multivariate analysis. Regarding OS, a multiple Cox regression model with a step-wise backward procedure for deleting factors above the cut-off significance level of 0.05 was used, while for factors associated with achievement of response a multinominal logistic regression analysis was applied. For all analyses, a *p*-value < 0.05 was considered to be statistically significant. Statistical analyses were performed using GraphPad Prism^®^ 5.01 (GraphPad Software Inc., La Jolla, CA, USA) and SPSS Statistics for Windows (SPSS Inc. Chicago, IL, USA). Data were locked for this retrospective analysis on 1 July 2019.

## 5. Conclusions

Taking into account the limitations of a retrospective analysis, our data show that the combination of Aza and DLI as first salvage therapy for post-transplant AML or MDS relapse achieves meaningful response rates, thus facilitating substantial long-term survival. Both response and overall survival are easily predictable using the proposed scoring systems, showing that those patients receiving treatment at a stage of low disease burden especially seem to benefit, whereas in the patient subgroup of MDS and sAML the pretransplant strategy also seems to impact the success of post-transplant salvage therapy with Aza and DLI.

## Figures and Tables

**Figure 1 cancers-12-02255-f001:**
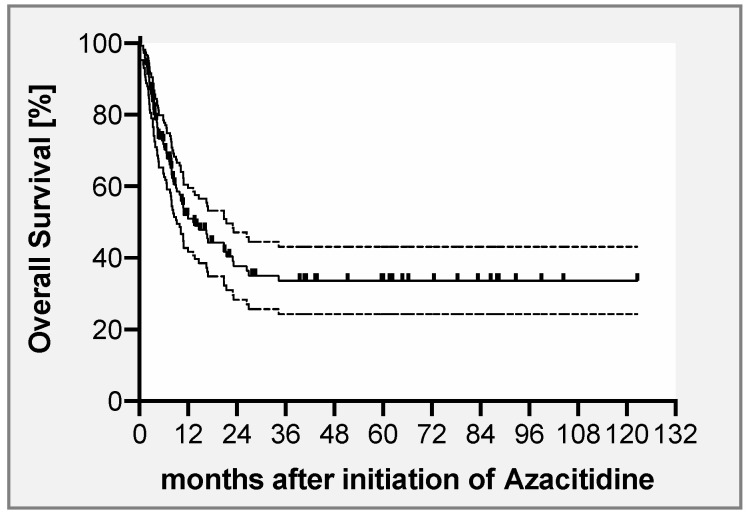
Overall survival (OS) after treatment with azacitidine ± DLI for relapse of myeloid neoplasms. OS of the study population comprising 151 patients was 38% (±9%) at 2 years.

**Figure 2 cancers-12-02255-f002:**
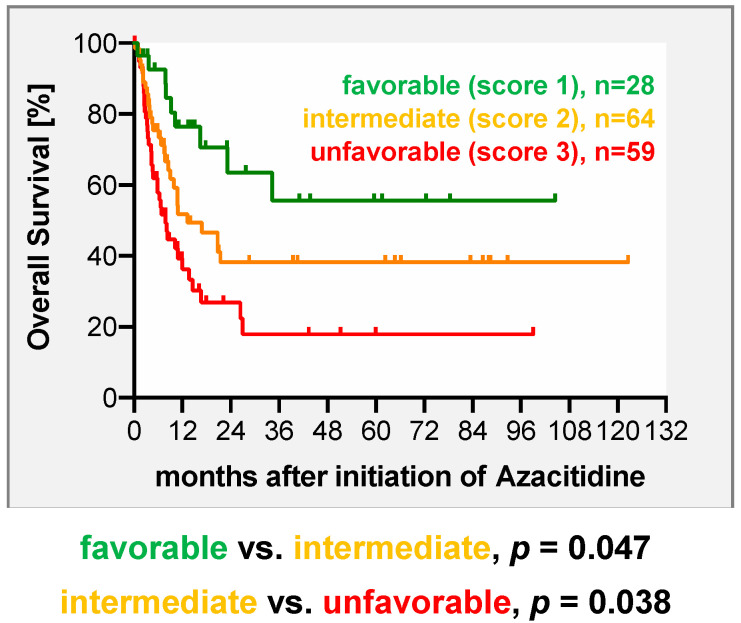
Overall survival (OS) after treatment with azacitidine ± DLI in 151 patients based on a scoring system including relapse type and time between allo-HSCT and diagnosis of relapse. One point was assigned for molecular relapse, while two points were given for diagnosis of hematologic relapse. An interval between transplant and relapse of ≥6 months was assigned with zero points, while for relapse within the first 6 months after transplant one point was given. As such, the model enables a stratification of patients into three distinct risk categories (favorable = 1 point, intermediate = 2 points, unfavorable = 3 points).

**Figure 3 cancers-12-02255-f003:**
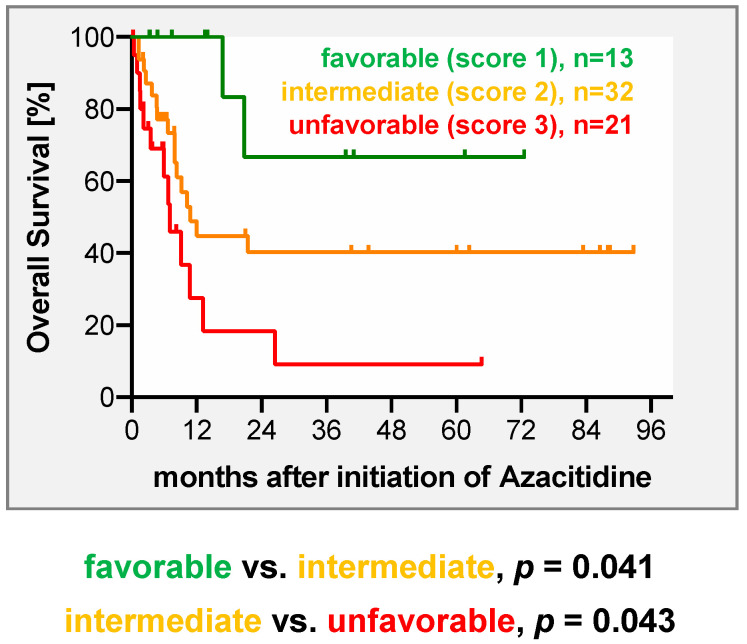
Overall survival (OS) after treatment with azacitidine ± DLI in 66 patients with MDS and sAML with 20–29% BM blasts (formerly RAEB-T) based on a scoring system including relapse type and pretransplant strategy. One point was assigned for molecular relapse, while two points were given for diagnosis of hematologic relapse. Upfront transplantation was assigned zero points, while for application of HMA or intensive CTX prior transplant one point was given. As such, the model enables a stratification of patients into three distinct risk categories (favorable = 1 point, intermediate = 2 points, unfavorable = 3 points)

**Table 1 cancers-12-02255-t001:** Patients demographics and clinical characteristics.

Characteristic	No.	%
**No. of patients**	151	100
**Age, median (range), y**	54 (19–71)	
**Gender** Male Female	83 68	55 45
**MDS/MPN, WHO 2016 *** MDS-MLD MDS-RS MDS-EB1 MDS-EB2 MDS-u CMML MPN (ET, CML)	6 2 9 30 2 10 2	4 1 6 20 1 7 1
**AML, WHO 2016 *** AML with recurrent genetic abnormalities AML with MDS-related changes Therapy-related AML NOS Myeloid sarcoma Missing	8 42 1 36 1 2	5 28 1 24 1 1
**Karyotype** Normal Aberrant Complex Monosomal Missing	58 86 40 7 7	38 57 47 8 5
**ELN Genetic Risk ^†^** Favorable Intermediate Adverse Missing	8 44 37 1	9 49 41 1
**Cytogenetic Risk ^‡^** Very good Good Intermediate Poor Very poor Missing	0 24 5 13 5 2	0 49 10 27 10 4
**Disease Status at allo-HSCT ^§/†^** Untreated Treated Remission CR1 CR2 PR No Remission Primary refractory Relapse Missing	43 104 50 45 5 15 39 19 20 4	28 69 48 90 10 14 38 49 51 3
**Conditioning ^||^** Standard-dose Dose-reduced	54 97	36 64
**Donor Type** Matched related Matched unrelated Mismatched unrelated Haploidentical	28 85 34 4	19 56 23 3
**Graft source** PBSC BM	150 1	99 1

Note: * according to Arber et al. [18]; ^†^ according to ELN-criteria, Döhner et al. [19] (regarding AML patients, n = 90); ^‡^ according to IPSS-R, Greenberg et al. [20] (regarding MDS patients, n = 49); ^§^ according to Cheson et al. [21]; ^||^ according to Bacigalupo et al. [22]; allo-HSCT, allogeneic hematopoietic stem cell transplantation; AML, acute myeloid leukemia; BM, bone marrow; CML, chronic myeloid leukemia; CMML, chronic myelomonocytic leukemia; CR1, first complete remission; CR2, second complete remission; ELN, European Leukemia Net; ET, essential thrombocythemia; IPSS-R, revised international prognostic scoring system; MDS, myelodysplastic syndrome; EB1, excess blasts 1; EB2, excess blasts 2; MDS-u, myelodysplastic syndrome-unclassifiable; MLD, multilineage dysplasia; MPN, myeloproliferative neoplasm; NOS, not otherwise specified; no., number; PBSC, peripheral blood stem cells; PR, partial remission; RS, ring sideroblasts; WHO, World Health Organization; y, years.

**Table 2 cancers-12-02255-t002:** Relapse characteristics.

Relapse Characteristics	All	Molecular Relapse	Hematologic Relapse
**No.**	151	59	92
**Time to relapse, median (range), months**	4.8 (0.9–211)	5.1 (0.9–211)	4.3 (0.9–110)
**WBC, median** **(range), ×10^9^/L**	3.7 (0.3–48.6)	3.8 (0.3–20.2)	3.5 (0.6–48.6)
**PB Blasts, median** **(range), %**	0 (0–71)	0 (0)	0 (0–71)
**BM Blasts, median** **(range), %**	7 (0–92)	2.5 (0–5)	21.5 (1–92)
**Median Hb, median** **(range), g/dL**	11.4 (7.2–17)	11.8 (7.4–15.7)	11.3 (7.2–17)
**Platelets, median** **(range), ×10^9^/L**	95 (6–800)	128 (18–800)	60 (6–205)
**PB chimerism, median** **(range), %**	86.3 (0–100)	97.8 (20.6–100)	77.1 (0–100)
**BM chimerism, median** **(range), %**	87.3 (1.2–100)	96.7 (8.3–100)	66.7 (1.2–99.7)
**Extramedullary relapse** isolated with systemic relapse	0 5	0 1	0 4
**aGvHD * before relapse** yes no ≥grade 3 **cGvHD ^†^ before relapse** yes no severe	50 (33%) 101 (67%) 12 (8%) 14 (9%) 137 (91%) 0	17 42 2 4 55 0	33 59 10 10 82 0
**Immunosuppression at Aza Start** yes no missing	61 (40%) 64 (42%) 26 (17%)	30 26 3	31 38 23
**aGvHD * after treatment with Aza +/− DLI** yes no ≥grade 3 **cGvHD ^†^ after treatment with Aza +/− DLI** yes no severe	63 (42%) 88 (58%) 16 (11%) 40 (26%) 111 (74%) 5 (3%)	25 34 9 20 39 3	38 54 7 20 72 2

Note: * modified according to Glucksberg et al. [23]; ^†^ according to Filipovich et al. [24]; aGvHD, acute graft-versus-host disease; Aza, Azacitidine; BM, bone marrow; cGvHD, chronic graft-versus-host disease; d, days; DLI, donor lymphocyte infusion; Hb, hemoglobin; PB, peripheral blood; WBC, white blood cells.

**Table 3 cancers-12-02255-t003:** Impacts of clinical parameters on response and outcome after treatment with azacitidine for posttransplant AML and MDS relapse—univariate and multivariate Analysis (n = 151).

Variable	Overall Survival			Response		
	2-y OS after Initiation of Aza (%)	*p*		CR Rate after Initiation of Aza (%)	*p*	
		Uni-variate	Multi-variate		Uni-variate	Multi-variate
**Age** ≥60 y <60 y	32.7 ± 8 41.8 ± 6	0.16	-	33 56	0.13	ns
**Gender** Female Male	27.9 ± 7 45.5 ± 8	0.07	ns	35 46	0.24	-
**Diagnosis** AML MDS	29.3 ± 6 49.4 ± 8	0.03	ns	41 47	0.59	-
**Karyotype** Abnormal Normal	35.1 ± 6 44.7 ± 8	0.28	-	35 48	0.12	ns
**Karyotype** Complex Not complex	26.1 ± 9 41.2 ± 6	0.16	ns	28 43	0.09	ns
**Molecular/genetic risk *** High Low/int	42.6 ± 7 27.5 ± 8	0.17	-	34 46	0.21	-
**Disease status at transplant**						
No CR	38.5 ± 6	0.3	-	36	0.08	ns
CR	45.8 ± 9			52		
**Donor** Related Unrelated	37.7 ± 11 39.4 ± 5	0.84	-	44 40	0.84	-
**HLA-Match** Mismatched Matched	36.2 ± 9 38 ± 6	0.51	-	29 45	0.09	ns
**Conditioning ^||^** RIC Standard dose	30.1 ± 6 51.2 ± 8	0.12	ns	36 50	0.12	ns
**Type of relapse** Hematologic Molecular	29 ± 6 54.6 ± 9	0.0004	0.002	28 61	<0.0001	0.0001
**Time until relapse** <6 months ≥6 months	31.4 ± 6 50.5 ± 8	0.007	0.026	33 52	0.02	ns
**BM blasts at relapse** >7% (median) ≤7%	32 ± 7 46.8 ± 7	0.001	ns	26 56	0.0005	ns

Note: * for MDS IPSS-R risk cytogenetics [20] were applied and very good, good and intermediate risk categories were assigned to low/intermediate group, whereas poor and very poor risk categories were included into high risk group; for AML ELN cytogenic risk classification (19) was applied and favorable and intermediate risk categories were assigned to low/intermediate risk group, whereas adverse risk category was included into high risk group; ^||^ according to Bacigalupo et al. [22]; - indicates that the respective parameter was not included into multivariate model; AML, acute myeloid leukemia; Aza, Azacitidine; BM, bone marrow; CR, complete remission; HLA, human leukocyte antigen; int, intermediate; MDS, myelodysplastic syndrome; ns, no statistical significance; OS, overall survival; RFS, relapse-free survival; RIC, reduced intensity conditioning; y, years.

**Table 4 cancers-12-02255-t004:** Two-year overall survival and response rate of 151 patients with MDS, sAML, and de novo AML treated with azacitidine ± DLI for relapse after allo-HSCT according to the proposed scoring system.

Risk Score/Group	Response Rate (CR) after Azacitidine	2-y OS Rate after Azacitidine [±SEM]
**1 (n = 28)**	71%	64% ± 11%
**2 (n = 64)**	39%	38% ± 8%
**3 (n = 59)**	29%	27% ± 7%
	*p =* 0.0007	*p* = 0.0012

Note: allo-HSCT, allogeneic hematopoietic stem cell transplantation; AML, acute myeloid leukemia; BM, bone marrow; CR, complete remission; DLI, donor lymphocyte infusion; MDS, myelodysplastic syndrome; OS, overall survival; sAML, secondary AML; SEM, standard error of the mean; y, year.

**Table 5 cancers-12-02255-t005:** Two-year overall survival and response rates of 66 patients with MDS and sAML with 20–29% BM blasts treated with azacitidine ± DLI for relapse after allo-HSCT according to the proposed scoring system.

Risk Score/Group	Response Rate (CR) after Azacitidine	2-y OS Rate after Azacitidine [±SEM]
**1 (n = 13)**	69%	67% ± 19%
**2 (n = 32)**	50%	38% ± 10%
**3 (n = 21)**	10%	18% ± 11%
	*p =* 0.0009	*p* = 0.0023

Note: allo-HSCT, allogeneic hematopoietic stem cell transplantation; AML, acute myeloid leukemia; BM, bone marrow; CR, complete remission; DLI, donor lymphocyte infusion; MDS, myelodysplastic syndrome; OS, overall survival; sAML, secondary AML; SEM, standard error of the mean; y, year.

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
