# Peer review of "Prediction of Response and Survival Following Treatment with Azacitidine for Relapse of Acute Myeloid Leukemia and Myelodysplastic Syndromes after Allogeneic Hematopoietic Stem Cell Transplantation"

_cancers, 2020, doi:10.3390/cancers12082255_

Round 1

Reviewer 1 Report

The present retrospective analysis showed suggestive results in the treatment of refractory disease such as relapse after allogenic transplant in myeloid disease including MDS and AML.  Authors build  score(s) which predict outcome of those setting. It may suggest important point of  view.

However, to conclude MRD-guided intervention with Aza+DLI after upfront transplantation without pre-transplant therapy for myeloid diseases is superior compared to reported strategy of SCT which employ pre-transplant Aza for those patients, additional explanation will be helpful.

  • Difference of outcomes between patient who received DLI and did not receive DLI, because treated cycles of aza for relapsed patient is highly variable, suggesting  role(s) of DLI is important.

  • Lines 301-303 in p12 is difficult to be understood. because the present study is retrospective, analysis including information of karyotype, which is well-known adverse prognostic factors in MDS, should be possible.

Reviewer 2 Report

Rautenberg et al. reviewed their results of azacitidine for relapse of AML or MDS following allogeneic transplantion. They identify 3 subgroups with different probability of response. The article is well written and complete.

I have few comments or questions:

  1. The techniques for identification of molecular relapse of most AML and MDS remains an area of controversy. It is uncertain whether various techniques used (chimerism, flow cytometry or NBS) are comparable. In the paper, molecular relapse is defined as a “decrease of donor chimerism to less than 95% measured in unsorted BM mononuclear cells”. How frequently BM chimerism was assessed (when was it firstly studied and how frequently was is it repeated ?). Did the authors used BM CD34+ cell chimerism as well ?

In addition, “increase in PB WT1-mRNA level >50 copies per 104 ABL copies” was also used. Was this latter technique use as a complementary method or in addition to the other methods for confirmation ?

  1. Upfront transplantation is proposed for some patients with advanced MDS or AML.       For those patients, what was the conditioning regimen ? Sequential ?

Reviewer 3 Report

The manuscript by Rautenberg et al deals with a major issue in the field of AML/MDS. In particular, the management of relapse after transplant has no SOC and studies addressing the predictive factors which may help the physicians to choose the best strategy are highly warranted. For this reason, I consider the paper of great interest. The study is very well-designed and the conclusions are in line with the results. I have no major questions for the authors. Albeit included in ELN classification, which is appropriately provided, no data regarding mutational status are presented. In particular, since FLT3 inhibitors may compete with AZA + DLI in FLT3mut patients, I was asking whether a specific note about this category of patients is not worth providing.
